# Effect of Hygrothermal Ageing on the Mechanical and Fire Properties of a Flame Retardant Flax Fiber/Epoxy Composite

**DOI:** 10.3390/polym14193962

**Published:** 2022-09-22

**Authors:** Charlotte Campana, Romain Léger, Rodolphe Sonnier, Patrick Ienny, Laurent Ferry

**Affiliations:** 1Polymers Composites and Hybrids (PCH), IMT Mines Alès, F-30319 Alès, France; 2LMGC, IMT Mines Ales, Montpellier University, CNRS, F-30319 Alès, France

**Keywords:** biocomposite, flax fibers, flame retardant, hygrothermal ageing, fire behavior, mechanical properties

## Abstract

In engineering applications, natural fiber composites must comply with fire requirements including the use of flame retardant. Furthermore, biocomposites are known to be water sensitive. Whether flame retardants affect the water sensitivity and whether water absorption affects the fire behavior and the mechanical performance of biocomposites are the two main topics addressed in this work. In this study, a flax fiber/epoxy composite flame retardant with 9,10-dihydro-9-oxa-10-phosphaphenanthrene-10-oxide (DOPO) or aluminum diethyl phosphinate (AlPi) was aged in humid atmosphere or by immersion in water. Water absorption kinetics revealed that DOPO induces an increase in equilibrium water content by approximately a factor of 2 due to its intrinsic hygroscopicity and/or its plasticizing effect on the epoxy matrix. In contrast, AlPi does not significantly change the water sensitivity of the biocomposite. Mechanical testing highlighted that, whatever the FR, the evolution of mechanical properties with ageing is governed by the moisture content. The drop of elastic modulus was attributed to a decrease in fiber rigidity due to plasticization, while the increase in tensile strength was assigned to an increase in fiber/matrix friction due to fiber swelling. As regards flame retardancy, only the highest water contents modified the fire behavior. For the AlPi containing biocomposite, the water release resulted in an increase by 50% of the time to ignition, while for the DOPO flame retardant biocomposite the water release was mainly postponed after ignition.

## 1. Introduction

Over the last two decades, natural-fiber-reinforced composites have aroused keen interest for engineering applications [1,2]. The main sectors that have promoted these materials are the automotive sector (seat backs, door cladding, parcel shelves, etc.) [3] and the construction sector (door, window frame, wall panels, etc.) [3,4,5]. The railway sector and the aircraft industry have also utilized biocomposites for seats, interior paneling, flooring and decking [6,7]. Natural fiber composites obviously compete with existing synthetic fiber composites (glass, carbon, Kevlar, etc.). In the context of current environmental concerns, biocomposites stand out due to their low carbon footprint as well as their non-hazardous and cheap solutions. Moreover, some intrinsic properties of these materials such as non-abrasiveness and fatigue resistance are remarkable [8,9,10]. However synthetic fiber composites are often superior for their absolute mechanical performance and their low water sensitivity [11,12]. Among natural fibers, flax occupies a prominent position for several reasons. First, flax can be cultivated in many countries with temperate and humid climate. Its growing cycle is short, which makes it a cost-effective crop [13]. Flax fibers are fine and regular, and they can be easily spun into yarns. Moreover, there is a traditional know-how in flax weaving and textile production. There has been over the years considerable improvement in agricultural methods used for flax production that has had consequences on the fiber geometry and properties [14]. Additionally, a considerable amount of work has been dedicated to the modification of flax fibers with physical and chemical treatments in order to improve the fiber/matrix adhesion and consequently the mechanical performance [15,16,17,18]. Epoxies are often used as a host matrix for biocomposites. This is justified by their mild processing temperature compatible with natural fibers, the large range of available processing techniques and the possibility of making this type of matrix partially bio-based.

Nevertheless, the use of natural fibers in composites shows also some weak points that have to be overcome or at least considered. One of the major drawbacks of lignocellulosic fibers related to their hydrophilic nature is their moisture sensitivity. Celino et al. studied the water sorption kinetics of several natural fibers and highlighted that for flax fibers the moisture uptake is approximately 62.5% when immersed in water, while it is approximately 12% in humid atmosphere [19]. When used as reinforcement in composite, flax fibers provoke an additional water uptake that significantly influences the mechanical behavior of the composite. Bergès et al. showed a decrease of 20% of the dynamic elastic modulus and an increase of 50% of the damping ratio for an epoxy/flax fiber composite after hygrothermal ageing at a temperature of 70 °C and relative humidity of 85% [20]. Cadu et al. mentioned that hygrothermal ageing induced fiber/fiber and fiber/matrix debonding as well as the plasticization of fibers responsible for the drop of elastic modulus, tensile stress and strain of their epoxy/flax fiber composite [21]. Working on the same type of biocomposite, Scida et al. also invoked other mechanisms such as flax fiber reorientation and matrix plasticization occurring during humid ageing [22]. These detrimental effects are likely to be attenuated by chemical treatments of the natural fibers as evidenced by Alix et al. [23,24].

Another drawback of natural fibers with regard to glass or carbon fibers is their flammability [25]. Their presence in composites modifies the thermal inertia and leads to a larger amount of fuel compared to inert reinforcement [26]. These changes generally lead to a decrease in the time to ignition and an increase in the total heat release. Some positive effects can nonetheless be observed, such as the formation of a char yield that can to some extent protect the underlying polymer and reduce the decomposition rate [27]. To fulfill specifications required for applications in industrial sectors such as transportation and building, biocomposites have to be flame retardant. The fireproofing of these materials can be achieved either by incorporating flame retardant in the matrix or by grafting or depositing appropriate molecules at the surface of the natural fibers. During the last decade, considerable effort has been made to modify the flax fiber surface by grafting or adsorption of phosphorous species. It was shown that, at equivalent phosphorus content, surface modification is more efficient than matrix modification in improving the fire behavior of biocomposites. However, a major bottleneck lies in depositing a large amount of flame retardant at the fiber surface [28]. As regards the flame retardancy of the epoxy matrix, many solutions are available [29]. In recent years, great effort has been made in the use of phosphorous compounds, especially DOPO derivatives and aluminum phosphinates [30]. Wang et al. evidenced that DOPO can react with epoxy and thus may be considered as a reactive flame retardant [31]. Liu et al. showed that DOPO acts both in the gaseous and condensed phases [32]. It promotes epoxy charring and acts as a smoke suppressant. Regarding aluminum phosphinate, Zhong et al. highlighted that aluminum diethyl phosphinate could reach an LOI of 28.5% and V0 UL94 ranking [33]. Liu et al. demonstrated that AlPi mainly acts in the condensed phase by increasing the char yield [34].

Considering the two aforementioned weak points of natural fiber composites, a corollary concern is the ageing of flame retardant biocomposites. From a general point of view, the durability of FR materials is rarely considered. The pioneering work of Clough deserves to be mentioned [35,36]. This researcher studied the flammability after radiation-thermal ageing of halogenated FR-containing formulations used for cables in nuclear plant applications. There is a great variety of ageing conditions including thermal, UV radiation, humidity, weathering (i.e., a combination of the three previous conditions), immersion, contact with organic solvent, ionizing (i.e., γ) radiation and electrical and thermal shocking. There are no general tendencies regarding the influence of ageing on flammability [37]. In some cases, ageing leads to a dramatic decrease in flame retardant properties [38]. However, many other systems appear to be resistant to ageing [39], and in some cases an improvement of flame retardant properties was noted [40]. Hence, the change in fire properties greatly depends on the FR system and the specific ageing conditions [41].

The most studied phosphorus-based FR is certainly ammonium polyphosphate (APP). It is used as an acid source in charring and intumescent systems, in combination with a char source (pentaerythritol or polymer), a blowing agent (melamine) and/or a mineral filler (char reinforcement). Lesaffre et al. studied the influence of different conditions (UV exposure, temperature and moisture) on the flammability of a PLA flame retardant with melamine, APP and organo-modified montmorillonite [42,43]. Even if PLA hydrolysis was observed (and enhanced in presence of FR fillers), the fire performance were improved due to the concentration of FR at the surface during ageing and also due to the dripping promoted by a decrease in molecular weight. APP-based FR systems are not durable in water immersion. Mangin et al. showed that phosphorus from APP was almost fully removed from a flame retardant PLA matrix after 3 weeks of ageing in water at 70 °C [44,45]. Therefore, the fire behavior measured in a cone calorimeter worsened. Jimenez et al. studied an FR epoxy coating deposited as a protective layer on steel [38]. The intumescent FR system contained APP, melamine and titanium dioxide. Few changes were observed after 1 month in distilled water. However, after one month in salt water, almost all phosphorus was removed from the coating. APP was transformed into much more soluble sodium polyphosphates. These species were easily removed from the coating. Melamine was also affected. Therefore, no more expansion was observed during exposition to a flame burner, evidencing the total loss of flame retardant properties. Encapsulation of APP was proposed to limit the degradation and preserve the fire performance [41,46,47,48]. Nevertheless, to date, few data are available on the influence of ageing on other common phosphorus FRs, especially organic ones such as DOPO and aluminum phosphinate.

The present study investigated the hygrothermal ageing of a flame retardant epoxy/flax fiber biocomposite containing DOPO or diethyl aluminum phosphinate. The novelty of this work lies in the attempt to investigate interactions between flame retardancy and humid ageing resistance in natural fiber composites. The first part outlines the study of the influence of FRs on the water sorption kinetics. In the second part, the effects of water sorption on the mechanical and fire properties are addressed.

## 2. Materials and Methods

### 2.1. Materials

The epoxy resin DER 332 (Ep) was provided by Dow Chemicals (Midland, TX, USA) with an epoxy equivalent weight of 170 g/eq. Isophorone diamine (IPDA) from Sigma-Aldrich (St. Louis, MO, USA) with a functionality of 4 was used as a hardener. The mixing of resin and hardener was carried out at a stoichiometric epoxy/amine ratio of 80 wt% DER 332 and 20 wt% IPDA. Quasi-unidirectional flax fabric UD 360 (Fl) was supplied by Fibres Recherche Développement (Troyes, France). Its areal weight is 360 g/m^2^ (weft: 330 g/m^2^; warp: 30 g/m^2^), and its thickness is 0.4 mm.

Two phosphorous compounds were used as flame retardants (FRs). Aluminum diethyl phosphinate (AlPi) was purchased from Clariant under the trade name Exolit OP 930. AlPi is a fine white powder that decomposes over 300 °C. 9,10-Dihydro-9-oxa-10-phosphaphenanthrene-10-oxide (DOPO) was purchased from Sigma-Aldrich. DOPO is a white powder that melts between 116 and 120 °C. The phosphorus contents of AlPi and DOPO are 23.7 and 14.4 wt%, respectively.

The composites were manufactured using a vacuum infusion process in controlled atmosphere (50% RH and 23 °C). The resin was heated at 40 °C to obtain a viscosity (of 120 mPa.s) suitable for the vacuum infusion process and then mixed with the hardener for 3 min at 400 rpm before the infusion. For fireproof composition, flame retardants were mixed with the pre-polymer before adding the hardener. The flame retardant content was adjusted so that the phosphorus content was 2 wt% with respect to the polymer. This leads to FR contents of 9.5 and 13.9 wt% for AlPi and DOPO, respectively. AlPi was mixed with the pre-polymer at room temperature for 1 h at 1200 rpm and then degassed for 50 min at 40 °C under vacuum. DOPO was mixed with the prepolymer at 120 °C for 1 h at 600 rpm.

Four plies of fabrics (300 × 300 mm^2^) oriented in the weft direction were infused at a constant pressure of 100 mbar for 30 min. The composite was then cured at 80 °C for 24 h. No post-curing treatment was performed to avoid any degradation of mechanical performance as highlighted in a previous paper [49]. The resulting composites contained 30 vol% of fibers. Composite plates (300 × 300 × 3 mm^3^) were then cut into samples of various sizes depending on the test to be performed and stored at 23 °C and 50 RH% according to ISO 291 and ASTM D618. The different composites prepared in this study are listed in Table 1.

### 2.2. Methods

#### 2.2.1. Ageing

Hygrothermal ageing was performed in a Weiss climate test chamber. Tests were mainly carried out at 70 °C and relative humidity (RH) of 50%, 65% and 85% (i.e., water activity a_w_ of 0.5, 0.65 and 0.85). These conditions were chosen to accelerate ageing without modifying the mechanisms and because data were already available for comparison [20]. Additional tests were performed at 90 °C/85 RH% for EpFl-AlPi and 40 °C/85 RH% for EpFl-DOPO. These conditions are justified in a later section. Ageing was performed for up to 30 days. Moreover, as a comparison, hydrothermal ageing was also carried out by immersing samples in deionized water at 70 °C.

Water uptake was measured by weighing the samples periodically at room temperature using a precision balance. Relative water uptake $W_{t}\;$ was determined as follows:(1)$$W_{t}=\frac{m_{t}-m_{0}}{m_{0}}$$
where $m_{t}$ is the mass of the sample at ageing time *t* while $m_{0}$ is the initial mass. The results correspond to the mean value of at least 3 repeats.

Water sorption kinetics was analyzed considering Fick’s ideal diffusion. In this model, diffusion is supposed to be driven by water concentration gradient. In a one-dimensional case, at a constant temperature, diffusion in the direction normal to the surface is described as follows:(2)$$\frac{\partial C\left(x,t\right)}{\partial t}=D\left(\frac{\partial^{2}C\left(x,t\right)}{\partial x^{2}}\right)$$
where *C* is the water concentration, *t* the ageing time and *D* the diffusion coefficient, assumed to be a constant. For a slab of thickness *h*, with an initial moisture concentration constant within the sample and environment moisture concentration at the boundaries constant all over the test, the solution for *C(x,t)* was given by Shen et al. [50]. The water absorption weight can be calculated by integrating the water concentration over the slab thickness. The relative water uptake resulting from this integration is given by Equation (3).
(3)$$W_{t}=W_{\infty }\left[1-\frac{8}{\pi^{2}}\sum_{n=0}^{\infty }\frac{1}{{\left(2n+1\right)}^{2}}\exp\left[\frac{-{\left(2n+1\right)}^{2}D\pi^{2}}{h^{2}}t\right]\right]$$
where $W_{t}$ is the relative water uptake at time *t* and $W_{\infty }$ is the relative water uptake at saturation or equilibrium water content (EWC). In the following, *n* = 20 was used to fit the experimental results.

For short time moisture exposure, it has been shown that the relative water uptake varies linearly with $\sqrt{t}$ according to Equation (4):(4)$$W_{t}=W_{\infty }\frac{4}{h}\sqrt{\frac{D}{\pi }}\sqrt{t}$$

Therefore, the diffusion coefficient can be determined from the slope *k* of $W_{t}$ versus $\sqrt{t}$ curve according to Equation (5).
(5)$$D=\pi {\left(\frac{k\;h\;}{4\;W_{\infty }}\right)}^{2}$$

#### 2.2.2. Glass Transition

Dynamic mechanical analysis (DMA) was performed to assess the glass transition temperature of the composites, which was assimilated to the main relaxation temperature $T_{\alpha }$. Tests were carried out using the single cantilever bending mode at a frequency of 5 Hz with an imposed peak-to-peak displacement of 5 µm. Specimen dimensions were 44 × 10 × 3 mm^3^. $T_{\alpha }$ was determined at the peak of tan δ curve.

#### 2.2.3. Mechanical Properties

The elastic modulus of composites was determined using a non-destructive impulse excitation technique described in more detail in [51]. The same samples were tested after different ageing periods. Composites samples of dimensions 250 × 25 × 3 mm^3^ were excited using a shock hammer, and the induced vibrational response was recorded using an accelerometer. Two weights of 86 g were fixed to the specimen ends in order to shift the first “tension–compression” mode natural frequency in the measurement range of the accelerometer (100–900 Hz). Weights were supposed to be integral with the bar. The longitudinal elastic modulus *E* was determined using Equation (6):(6)$$E=\frac{\rho L^{2}f_{1}^{2}}{{\left(\frac{\beta_{1}}{\pi }\right)}^{2}}$$
with $\beta_{1}$ being the lowest positive solution of Equation (7):(7)$$\beta_{1}\tan\beta_{1}=\frac{m}{M}$$
where *M* is the total additional weight (172 g); *L*, *m* and *ρ* are the length, mass and specific density, respectively, of the specimen and $f_{1}$ is the first “tension–compression” mode natural frequency. The relative standard deviation of this test was assessed to be 1.4%.

Uniaxial tensile tests were performed on an MTS testing machine (model Criterion C45.105) equipped with a 100 kN load cell. The tests were performed at a constant displacement rate of 1 mm/min. The deformation was measured using an MTS laser extensometer (Model LX500) with a precision of 1 µm. Sample dimensions were 250 × 25 × 3 mm^3^ for the composites following the ISO 527 standard. Taking into account the bilinear behavior of composites at low strain, Young’s modulus was determined between 0.5 and 0.8% of strain according to a procedure defined in a previous paper [49]. Tensile strength as well as elongation at break of composites were also determined. Results correspond to the mean value of 4 repeats.

#### 2.2.4. Density

The specific density was measured using a Micromeritics Helium pycnometer (AccuPyc 1330 model). Results correspond to the mean value of 4 repeats.

#### 2.2.5. Fire Properties

The fire behavior of composites was assessed using a Fire Testing Technology (FTT) Cone Calorimeter. Tests were performed in accordance with the ISO 5660-1 standard. The specimens (100 × 100 mm section, 3 mm thickness) were horizontally exposed to an irradiance of 50 kW/m^2^. Time to ignition (TTI), peak of heat release rate (pHRR), total heat release (THR), residual mass and effective heat of combustion (EHC) were determined. Each composition was tested in triplicate. The measurement uncertainties of HRR are known to be around 10% with this apparatus.

#### 2.2.6. Microstructure

A scanning electron microscope (FEI Quanta 200 SEM) equipped with an energy dispersive X-ray (EDX) spectrometer (Oxford INCA Energy 300) was used to study the morphology of composites after tensile tests. Observations were performed at an accelerating voltage of 15 kV and a working distance of 10 mm.

## 3. Results and Discussion

### 3.1. Water Sorption Kinetics

#### 3.1.1. Influence of Water Activity

Figure 1 shows the water sorption kinetics of EpFl, EpFl-AlPi and EpFl-DOPO at 70 °C for three values of relative humidity (50%, 65% and 85%). It can be observed that, whatever the materials and the conditions, the water uptake is linear with $\sqrt{t}$ in the early stage of ageing and reaches a saturation plateau for a longer duration. This behavior is characteristic of a Fickian diffusion process. For this reason, the kinetics curves were fitted by using the formalism previously described in Equations (4) and (5).

Figure 2 represents the equilibrium water content (EWC) in composites as a function of water activity a_w_ at 70 °C. EWC of biocomposites can be compared to the EWC of their components. At a_w_ = 0.85 and T = 70 °C, EWC of the non-flame retardant biocomposite (EpFl) is 2.55%, while it was measured to be 1.1% for the epoxy resin and approximately 13% for flax fibers [23,52]. This shows that EWC of EpFl is lower than predicted by a rule of mixtures of its components. This may be attributed to the fact that flax fiber swelling is hindered by the surrounding matrix in the composite, thus limiting the water uptake.

It is noteworthy that water sorption isotherms exhibit deviation from the linear behavior of Henry’s law, indicating that EWC is not only governed by water solubility in the biocomposite materials. In the literature, epoxy resins exhibit water sorption isotherms that deviated only slightly from Henry’s law [53,54]. In contrast, flax fibers exhibit a sorption isotherm with a strong increase in EWC at high water activity [23,52]. In our case, the curvature observed at high water activity may be related either to clustering phenomena (cluster of water molecules) or to the swelling of flax fibers [55,56]. Such behavior can be well described by the Park model [57] (Equation (8)), which corresponds to a multi-sorption model where a power law term accounts for the increase in EWC at high water activity:(8)$$W_{\infty }=K_{H}a_{w}+\frac{A_{L}b_{L}a_{W}}{1+b_{L}a_{W}}+K_{a}a_{w}^{n}$$

With *K_H_* Henry’s solubility, *A_L_* the Langmuir capacity constant, *b_L_* the Langmuir affinity and *K_a_* and n two constants describing clustering.

In our case, fitting experimental points with the Park model leads to the Henry and Langmuir terms being ignored, indicating that water sorption is mainly governed by clustering. Optimization gives *K_a_* values of 3.74, 4.57 and 7.63, while values of *n* are 2.1, 2.7 and 2.2 for EpFL, EpFl-AlPi and EpFL-DOPO, respectively.

When considering the flame retardant biocomposites, it can be observed that the presence of AlPi affects neither the kinetics of water sorption nor the EWC, which are similar to those of EpFl whatever the water activity. Therefore, *K_a_* and n for those materials are very similar. This is not really surprising since AlPi is claimed to be non-hygroscopic owing to the presence of three diethyl groups [58]. On the contrary, the presence of DOPO induces a significant increase in EWC, which is roughly twice that of EpFl, highlighted by a doubling of *K_a_*. There are two possible explanations for the increase in EWC. The first reason is the plasticizing effect of DOPO that decreases the glass transition of the epoxy resin by almost 30 °C (see Section 3.2). Plasticization of the epoxy network leads to two consequences likely to increase the water sorption ability: (i) a higher free volume in the polymer matrix and (ii) a decrease in polymer rigidity favorable to flax fiber swelling [56]. The second reason for EWC increase is the high polarity of the P-H bond in DOPO that may act as a hydrophilic site for water sorption [59].

The water sorption rate can be appreciated through the value of the diffusion coefficient D. D was determined for the different biocomposites and the different testing conditions using Equation (5). Figure 3 highlights that the diffusion coefficient increases with the increase in activity. This behavior was also observed in the literature by Broudin et al. in PA66 [60]. The authors mentioned that in the glassy state, D is almost constant at low water activity, while it increases at high water activity. This phenomenon was assigned to the plasticization of the polymer matrix by the water molecules as ageing progresses. In our case, a similar plasticizing effect was observed as demonstrated later. The presence of flame retardant (AlPi or DOPO) does not significantly change the value of the diffusion coefficient.

#### 3.1.2. Influence of Temperature

Temperature plays a dominant role in accelerated ageing tests. In order to assess the sensitivity to temperature of water sorption, two additional tests were performed at a water activity of 0.85. A test was performed on EpFl-DOPO at 40 °C. This temperature was chosen to obtain a similar Tg-T_ageing_ gap compared to the other biocomposites. Indeed, the glass transition of EpFl-DOPO was shown to be 30 °C lower than those of EpFl and EpFl-AlPi. A second test was performed at 90 °C on EpFl-AlPi in order to accelerate ageing while staying in the glassy state of epoxy. Figure 4 shows the sorption kinetics measured in these two tests. Regarding the test on EpFl-DOPO, it can be observed that EWC is lower at 40 °C compared to 70 °C. However, it must be underlined that saturation was probably not completely reached after 1200 h of ageing. This result is consistent with those obtained by Scida et al. on flax/epoxy composites [22] or by Choqueuse et al. on glass/epoxy composites [61]. In both cases, EWC increases with the increase in temperature. A similar behavior was also observed for pure polymers, i.e., PA66 and PET [60,62]. On the contrary, it can be noticed that for EpFl-AlPi, EWC is not modified by the increase in temperature. In this case, it is assumed that post-curing may occur during ageing at 90 °C, and this phenomenon may counter-balance the effect of plasticization by limiting the number of hydrophilic sites or limiting fiber swelling, and thus EWC remains constant. This effect is not related to the presence of AlPi and would surely be observed whatever the biocomposite.

Figure 4 highlights that, in both cases, increasing the ageing temperature leads to increase the water sorption rate. In the glassy state, the water diffusion coefficient is supposed to be dependent on temperature according to an Arrhenius law (Equation (9)), as long as water diffusion does not significantly modify the physical properties of the polymer [60,63]:(9)$$D=D_{0}\exp\left(\frac{-E_{a}}{RT}\right)$$
where $E_{a}$ is the activation energy for diffusion, $D_{0}$ the pre-exponential factor and R the gas constant. In Figure 5, ln *D* is plotted versus 1/T. Insofar as EpFl-AlPi and EpFl-DOPO exhibited a similar diffusion coefficient at 70 °C, a linear regression was made considering the four experimental values. An activation energy of 48.8 kJ/mol and a pre-exponential factor $D_{0}$ of 9.4 × 10^−3^ m^2^/s were determined. The activation energy is closed to the one determined by Barrie et al. [54] on epoxy resins, indicating that the matrix may govern the diffusion rate.

#### 3.1.3. Humid Atmosphere versus Immersion

The water sorption kinetics of biocomposites was also investigated in deionized liquid water at 70 °C. The results are presented in Figure 6 and compared to the kinetics obtained in water vapor at a_w_ = 0.85. It can be observed that water uptake also follows a Fickian behavior in the investigated range of time. Using Equation (5), the diffusion coefficient was calculated and found to be 4.8 × 10^−10^, 3.5 × 10^−10^ and 4.6 × 10^−10^ m^2^/s for EpFl, EpFl-AlPi and EpFl-DOPO, respectively. These values are slightly higher but close to those obtained at water activity of 0.85, consistent with that extrapolated in saturated humid atmosphere from Figure 3. As observed in water vapor, it is noteworthy that EpFl-DOPO absorbs a larger amount of water than EpFl and EpFl-AlPi. However, values of EWC for the three biocomposites are much higher than the ones predicted by the Park model in saturated humid atmosphere in Figure 2. This difference may be attributed to water clustering that would only occur when ageing is carried out by immersion.

### 3.2. Glass Transition

The glass transition temperature is an important property in hygrothermal ageing of polymers. Depending on the position of ageing temperature relative to Tg, the law governing the evolution of the water diffusion coefficient with temperature may be affected, changing from an Arrhenius law to a more complex expression involving the free volume theory [60]. Indeed, above Tg, the increase in free volume accelerates the diffusion rate. Moreover, water molecules are also good plasticizers that may depress Tg.

The glass transition of epoxy was determined by DMA for the various biocomposites having reached equilibrium after water sorption at 70 °C. Figure 7 shows the evolution of Tg as a function of EWC. Firstly, the graph highlights the significant plasticizing effect of DOPO since Tg of unaged EpFl-DOPO is around 30 °C lower than that of unaged EpFl. As previously mentioned, this effect is supposed to be one of the reasons for the higher water uptake of biocomposite containing DOPO. The plasticizing effect of DOPO in epoxy has already been reported by Schäfer et al. with a Tg depression of 24 °C by percent of phosphorus content [64]. On the contrary, EpFl-AlPi exhibits a glass transition temperature similar to EpFl. Secondly, Figure 7 also evidences the plasticizing effect of water molecules on epoxy resin since there is a decrease in Tg with the increase in water content. However, it must be underlined that this effect is very moderate since Tg depression is approximately 2 °C/%water content. This value is much lower than that mentioned by Broughton et al., i.e., 13.5 °C/%water content, in F922 epoxy [65]. It is assumed that, in our case, a large amount of water is absorbed by flax fibers or localized in clusters, and therefore the actual water content of the resin is lower than the mean value in the biocomposite.

In order to evaluate the effect of moisture within a polymer on its glass transition temperature, the Fox equation (Equation (10)) is generally used [62]:(10)$$\frac{1}{T_{g}}=\frac{w_{p}}{T_{g,p}}+\frac{w_{w}}{T_{g,w}}$$
with $w_{p}$ and $w_{w}$ the weight fractions of polymer and water, respectively. $T_{g,p}$ is the glass transition temperature of the polymer in the dry state. $T_{g,w}$ is the glass transition temperature of water taken equal to 110 K according to the literature [60]. Assuming that plasticization of the epoxy follows the Fox equation, the weight fraction of water in the matrix at equilibrium was determined by a reverse approach for experiments carried out at a_w_ = 0.85 and *T* = 70 °C. $w_{w}$ was found to be 0.5%, 0.4% and 0.8% for EpFl, EpFl-AlPi and EpFl-DOPO, respectively. These values are slightly lower but consistent with the value experimentally measured in the pure resin after ageing at a_w_ = 0.85 and *T* = 70 °C, i.e., 1.1%, confirming the non-homogeneous repartition of water in the biocomposite.

### 3.3. Mechanical Properties

The non-destructive impulse excitation technique enabled the dynamic longitudinal modulus to be measured at room temperature throughout the ageing period by briefly exiting the sample from the climate test chamber. Elastic moduli of unaged samples are presented in Table 2. These values are very close despite the lower glass transition temperature observed for EpFl-DOPO. It is assumed that room temperature is far from the glass transition temperature, and therefore all samples are on the glassy plateau.

In the following, for simple comparison, the relative modulus, defined as the ratio of the modulus of aged sample upon modulus of unaged sample, is used. Figure 8 shows the evolution of the relative dynamic modulus as a function of water content for all experiments performed at an ageing temperature of 70 °C. Samples are not necessarily at equilibrium since measurements are made throughout ageing. Despite this, it can be observed that the elastic modulus of biocomposites is mainly governed by the water content since all values fall roughly on the same curve. Elastic properties of samples aged at 40 or 90 °C follow the same tendency (not presented for clarity). The elastic modulus decreases linearly with the moisture content at a rate of approximately 3.7%/%water content. The same trend was found when measuring Young’s modulus with tensile tests (results not presented here). A similar evolution has been mentioned in the literature on flax/epoxy composites without quantifying this effect [20,66]. This drop may be assigned to the plasticization of epoxy resin or to the drop of flax fiber rigidity due to the plasticization of lignin and hemicellulose inside the fiber wall. Taking into account the low plasticization of the matrix as proved by Figure 7, the second phenomenon is probably predominant. Using a reverse approach, Masseteau et al. estimated that the flax fiber elastic modulus decreased by 17.8% when moisture content varied from 2 to 8% [66].

Tensile tests were performed on all series having reached equilibrium according to the procedure mentioned in the experimental part. Figure 9 illustrates the evolution of tensile strength with EWC after ageing at 70 °C. Even if standard deviation is sometimes high, it can be concluded that tensile strength tends to increase with the increase in moisture content. This phenomenon was assigned to the swelling of fibers during water sorption that generates friction at the matrix/fiber interface, thus enhancing the load required to break the sample [67]. Furthermore, it can be seen in Figure 9 that the tensile strength significantly decreases after immersion in liquid water. In this case, SEM observations reveal that the drop of mechanical resistance can be attributed to a degradation of the external wall of the natural fibers provoked by the high water uptake (see Figure 10). This phenomenon has already been observed by Li et al. for flax fiber/epoxy composite after immersion in water at 60 °C [68]. At high water content, the swelling of fibers creates internal stresses that are likely to damage their structure by delamination of the different walls. With long ageing time, water may also dissolve hemicelluloses and pectins that act as a binder in primary wall of fibers. The load can no longer be transmitted to cellulose fibrils, and consequently the tensile strength of the biocomposite decreases.

As regards elongation, Figure 11 shows that it increases with the increase in moisture content. This evolution is mainly governed by the decrease in elastic modulus as highlighted in Figure 8 which dominates the behavior over the increase in tensile strength. These results are consistent with those obtained by Munoz et al. that also observed an increase in tensile strain that they attributed to plasticization of both the resin and the flax fibers [67].

### 3.4. Fire Properties

The fire properties of biocomposites were assessed through cone calorimeter tests using an external heat flux of 50 kW/m^2^ corresponding to a developing fire scenario. Figure 12 presents the HRR curves of unaged samples. It can be observed that EpFl exhibits typical behavior of an intermediate thick non-charring sample according to the classification of Schartel and Hull [69]. HRR increases gradually in the first part of the test, and the peak occurs when the heat front reaches the sample back face. As heat cannot be removed due to the isolating support, the pyrolysis rate is increased, leading to the HRR peak [70]. Finally, HRR decreases due to fuel depletion. The incorporation of both flame retardants leads to very similar effects in flax epoxy composite. AlPi has almost no effect on the time to ignition of the biocomposite that remains similar to that of EpFl. DOPO induces a slight decrease in TTI that may be related to its effect on epoxy thermal stability. Wang and Lin observed that the onset of degradation of epoxy tends to decrease with the increase in DOPO content [31]. After ignition, a first peak is observed, and then HRR drops significantly due to the action of the flame retardants. Both FRs are known to act mainly in the gas phase as confirmed by the low increase in char residue and by the decrease in EHC. According to several authors [71,72], AlPi decomposes through the formation of diethylphosphinic acid, which is released in the gas phase, and aluminum phosphate, which remains in the condensed phase. This latter compound may account for a 5% increase in residue. DOPO was also reported to act mainly in the gas phase since its decomposition releases phosphorus-containing fragments that are likely to trap high energy radical H° or OH°, thus leading to flame inhibition [73]. As soon as FRs become active, a significant decrease in HRR down to 250–300 kW/m^2^ is observed. The slowdown of burning rate shifts the second peak related to heat feedback at the back face towards a longer duration. On the whole, FRs diminish the mean value of HRR and increase the burning period.

The effect of ageing on fire properties was first studied with the EpFl reference for samples having reached EWC. The results (see Table 3) indicate that ageing has only a slight influence on composite flammability since neither TTI nor the mean HRR values are significantly modified. Figure 13 reports the HRR curves of EpFl-AlPi samples aged at 70 °C in various conditions. It is noteworthy that all curves exhibit the same shape. The major effect of ageing is a shift of the curve towards a long duration when EWC is high (ageing by immersion in liquid water). This result contrasts with those obtained by Oztekin et al. [74] and Safranova et al. [75] who observed that polymers containing moisture exhibited shorter TTI. This phenomenon was assigned to the increase in the surface sample heating rate due to a strong bubbling when water is released. Bubbling generates a foam structure slowing down heat transfer and thus accelerating surface degradation rate. However, those results were obtained for thermoplastic polymers (PEEK, POM, PA66, PMMA, PC). Our results are close to those obtained by Le Lay and Gutierrez [76] for polyester and vinyl ester composites or those of Shi and Chew [77] for woods where an increase in TTI was mentioned in wet samples compared to the dry ones. In the case of EpFl-AlPi, it is assumed that a part of the incident heat flux is used for the endothermic vaporization of moisture contained in the biocomposite, thus postponing the time required to reach the ignition temperature. Once this effect of water fades away, the sample ignites and burns in a slightly shorter period of time than the dry sample does.

In the case of EpFl-DOPO samples, Figure 14 shows that the presence of moisture has no influence on TTI or the first peak of HRR, which remains almost constant. It is also noteworthy that THR remains unchanged. The main difference lies in the drop of HRR just after the first peak of HRR and a shift of the second peak of HRR to longer durations, approximately 40 s compared to the dry sample. This shift could be assigned to the endothermic release of water that slows down the decomposition rate of the material after ignition.

On the whole, the effect of ageing on the fire behavior of composites is only significant when the moisture content is sufficient (>3%). The effect consists mainly of a time lag of the production of fuel. However, two situations may be distinguished. In the case of EpFl-AlPi, the tested high moisture containing sample was obtained by immersion in liquid water. As previously mentioned, this kind of ageing leads to water clustering in addition to the other sorption mechanisms. When exposed to the radiant source, it is assumed that water contained in clusters may be released more easily and therefore postpone ignition and then the whole HRR curve. As regards EpFl-DOPO, the tested sample with a high moisture content was obtained after ageing at a_w_ = 0.85. In this type of ageing, it is supposed that water does form clusters, but is mainly sorbed on hydrophilic sites of fibers, matrix and DOPO. Hence, when exposed to the radiant source, water release would be more difficult and would occur mainly after ignition.

## 4. Conclusions

Among the two tested phosphorous flame retardants, only DOPO significantly modifies the water sorption of the epoxy/flax fiber biocomposite during hygrothermal ageing. Due to its hydrophilic nature, it almost doubled EWC while it plasticized the epoxy matrix. The analysis of sorption isotherms evidenced that water is mainly absorbed by flax fibers or clustered on hydrophilic sites of FR. The linear decrease in glass transition temperatures with the water content close to that of a pure epoxy confirms the non-homogeneous repartition of water within the biocomposites. The ageing temperature tends to increase the diffusion coefficient of water in the materials without significantly modifying EWC. During humid ageing, it was highlighted that the changes in mechanical performance are mainly driven by the moisture content and follow the same trend irrespective of the FR used. The elastic modulus decreases due the plasticization of flax fibers, while the tensile strength and elongation at break increase due the swelling of the fibers, which results in a higher friction with the epoxy matrix. Finally, it was observed that the fire behavior of aged biocomposites was altered only at high water content. In the case of EpFl-AlPi, it leads to an increase in the time to ignition due to the early vaporization of water. In the case of EpFl-DOPO, it is assumed that water is released over a broader time period, thus modifying the second part of the HRR curve.

## Figures and Tables

**Figure 1 polymers-14-03962-f001:**
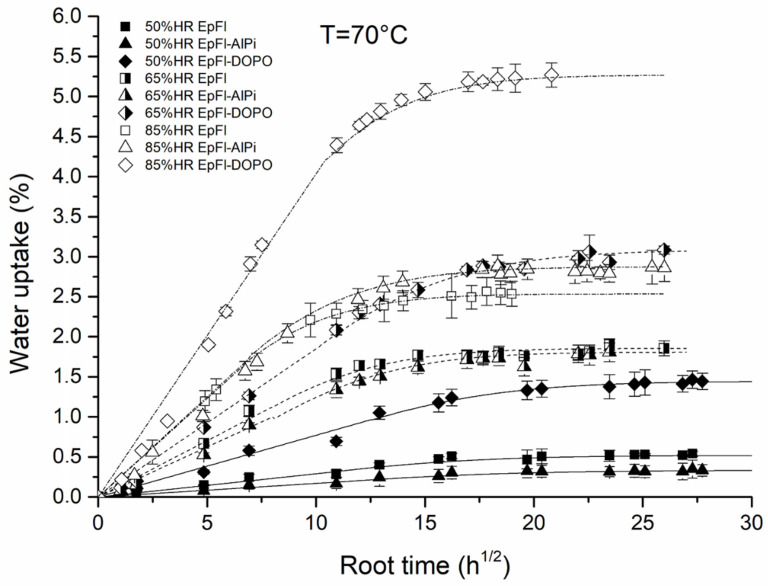
Water sorption kinetics at 70 °C in humid atmosphere for EpFl, EpFl-AlPi and EpFl-DOPO.

**Figure 2 polymers-14-03962-f002:**
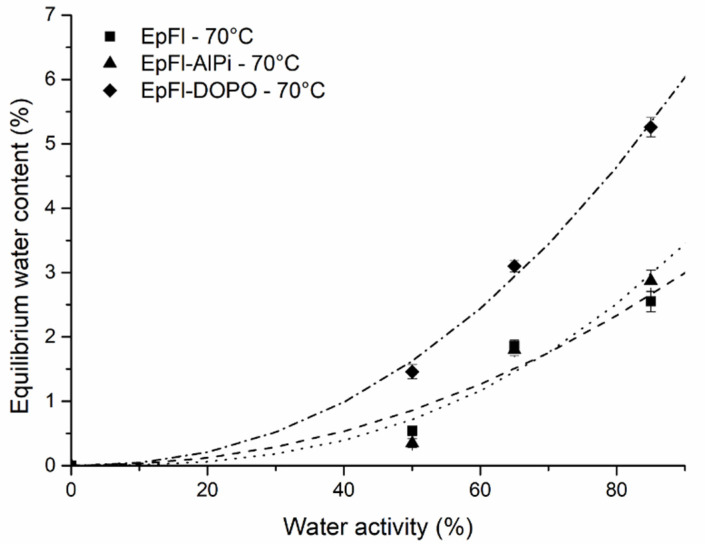
Equilibrium moisture content as a function of water activity at 70 °C.

**Figure 3 polymers-14-03962-f003:**
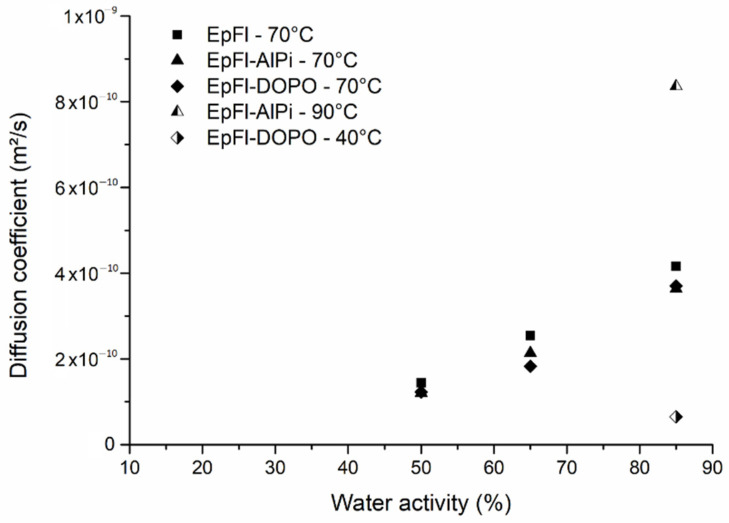
Equilibrium moisture content as a function of water activity at 70 °C.

**Figure 4 polymers-14-03962-f004:**
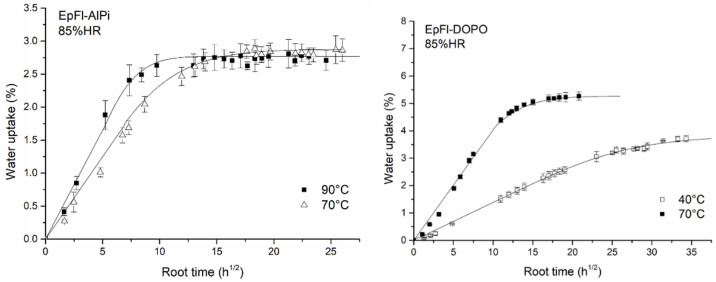
Water sorption kinetics of EpFl-AlPi at 90 °C and EpFl-DOPO at 40 °C.

**Figure 5 polymers-14-03962-f005:**
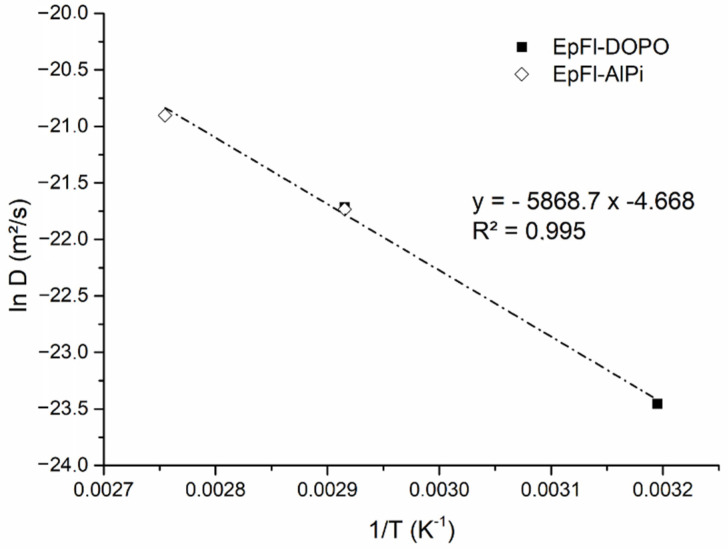
ln D versus 1/T curve for EpFl-AlPi and EpFl-DOPO for ageing at a_w_ = 0.85.

**Figure 6 polymers-14-03962-f006:**
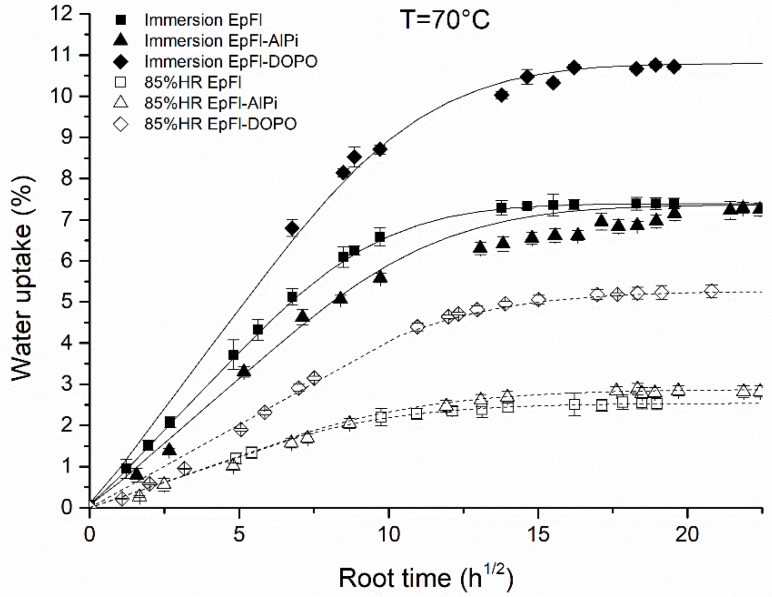
Water sorption kinetics of biocomposites at 70 °C in liquid water and water vapor.

**Figure 7 polymers-14-03962-f007:**
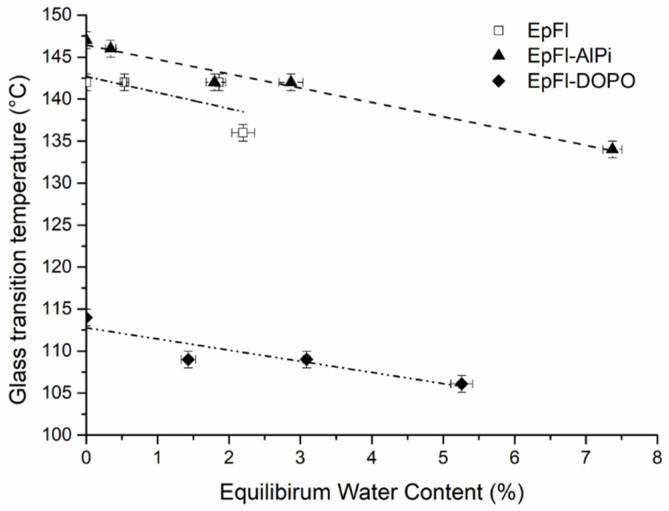
Glass transition temperature as a function of equilibrium water content.

**Figure 8 polymers-14-03962-f008:**
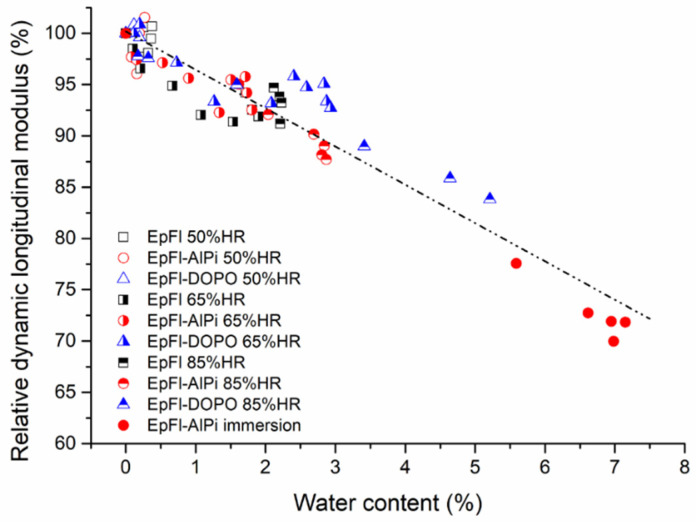
Relative dynamic longitudinal modulus versus water content for series aged at 70 °C.

**Figure 9 polymers-14-03962-f009:**
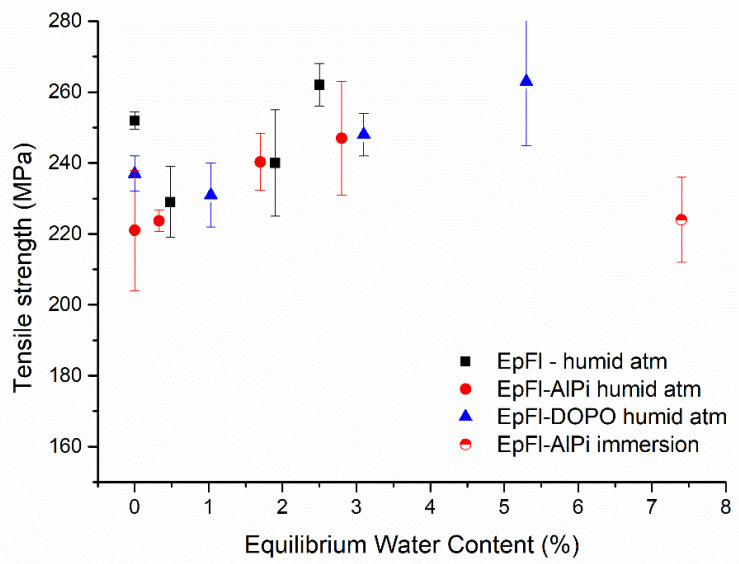
Tensile strength versus EWC for series aged at 70 °C.

**Figure 10 polymers-14-03962-f010:**
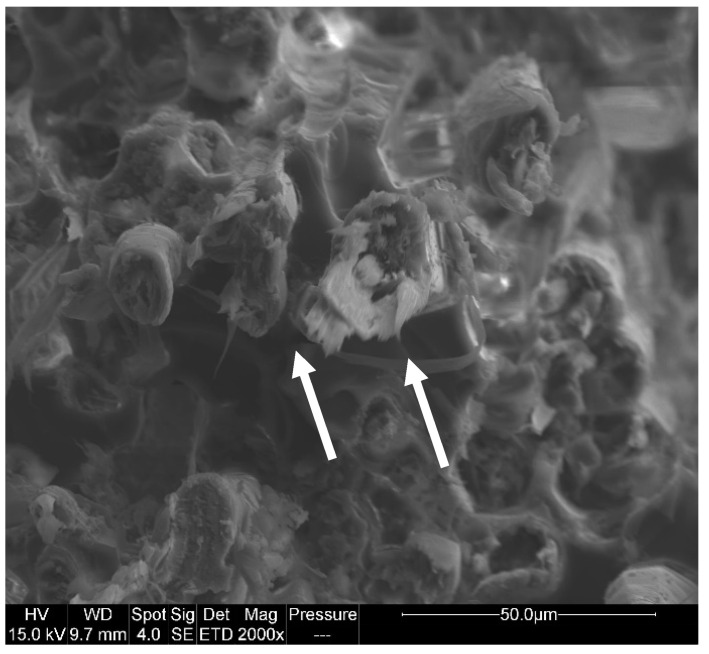
SEM picture of fracture surface of EpFl-AlPi after immersion in liquid water. Arrows indicate the tear of fiber external cell wall.

**Figure 11 polymers-14-03962-f011:**
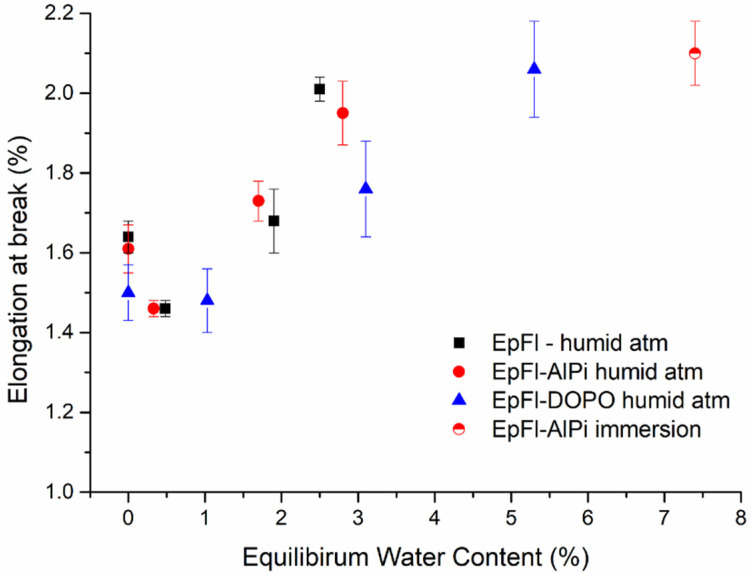
Elongation at break versus EWC for series aged at 70 °C.

**Figure 12 polymers-14-03962-f012:**
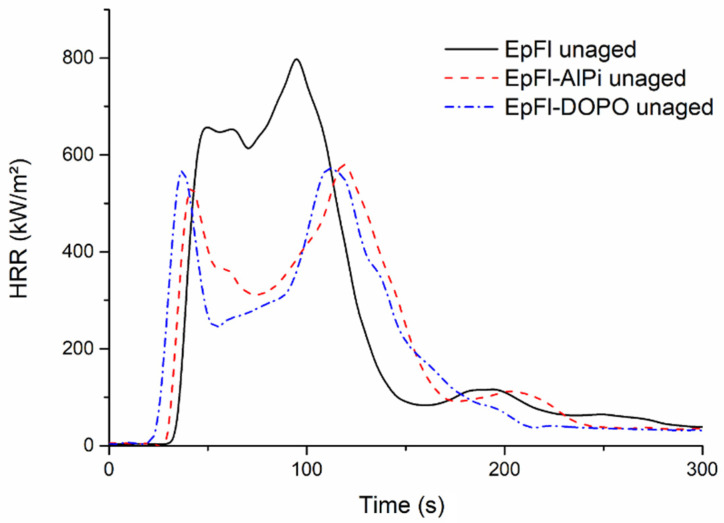
Cone calorimeter curves of unaged biocomposites.

**Figure 13 polymers-14-03962-f013:**
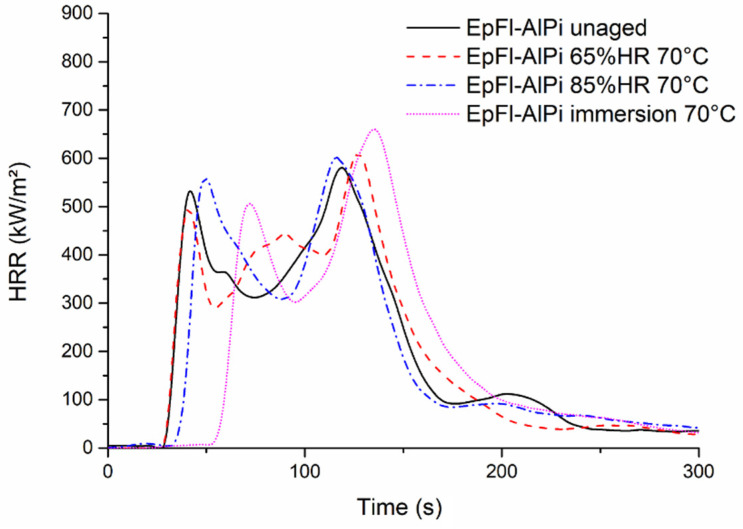
Cone calorimeter curves of EpFl-AlPi aged at 70 °C in different conditions.

**Figure 14 polymers-14-03962-f014:**
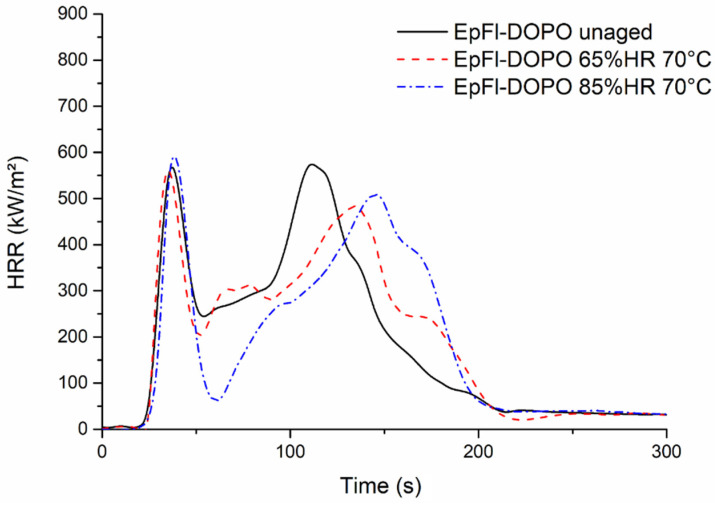
Cone calorimeter curves of EpFl-DOPO aged at 70 °C in different conditions.

**Table 1 polymers-14-03962-t001:** Composition of the different biocomposites and their Tg before ageing.

Designation	Matrix (vol%)	Flax (vol%)	FR (wt% of Matrix)	Tg (Dry) (°C)
EpFl	70	30	0	142
EpFl-AlPi	70	30	9.5	147
EpFl-DOPO	70	30	13.9	114

**Table 2 polymers-14-03962-t002:** Dynamic longitudinal modulus of unaged biocomposites.

	EpFl	EpFl-AlPi	EpFl-DOPO
Dynamic longitudinal modulus (GPa)	22.8 ± 0.21	22.7 ± 0.05	23.7 ± 0.21

**Table 3 polymers-14-03962-t003:** Cone calorimeter data for the various composites after ageing.

		EWC (wt%)	TTI (s)	pHRR1 (kW/m^2^)	tpHRR1 (s)	pHRR2 (kW/m^2^)	tpHRR2 (s)	Mean HRR (kW/m^2^)	Mean EHC (kJ/g)	THR (MJ/m^2^)	Residue (wt%)
EpFl	dry	0	38	682	50	826	100	262	20.8	75	1.7
	a_w_ = 0.65	1.9	37	614	45	754	115	259	19.5	72	2.6
	a_w_ = 0.85	2.5	37	733	58	778	113	270	20.6	77	2.4
EpFl-AlPi	dry	0	39	592	48	595	118	215	17.4	64	6.7
	a_w_ = 0.65	1.7	35	510	47	593	125	221	16.9	60	8.2
	a_w_ = 0.85	2.9	39	534	45	593	118	207	17.0	64	5.3
	immersion	7.4	57	490	67	661	137	220	16.9	59	8.1
EpFl-DOPO	dry	0	32	619	48	609	103	211	16.8	63	4.4
	a_w_ = 0.65	3.1	33	530	43	439	137	206	15.7	57	9.5
	a_w_ = 0.85	5.3	34	609	43	487	145	204	15.7	61	5.8

## Data Availability

Available data can be obtained from the corresponding author upon request.

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
