# Peer review of "Effect of Hygrothermal Ageing on the Mechanical and Fire Properties of a Flame Retardant Flax Fiber/Epoxy Composite"

_polymers, 2022, doi:10.3390/polym14193962_

Round 1

Reviewer 1 Report

The effect of hygrothermal ageing on the mechanical and fire properties of a flame retarded flax fibres/epoxy composite was investigated. A large number of data have been obtained and provided some basic data for supporting the engineering application of natural fiber composites. The following comments can further improve the quality of the paper.

1. The writing of the abstract should include some quantitative analysis on hygrothermal ageing and flame retardancy, etc. In addition, it is suggested to provide some application background of natural fiber epoxy composite composites at the beginning.

2. It is well known that the types of fibers mainly include natural fibers and synthetic fibers with different advantages. As far as synthetic fibers are concerned, they have very excellent mechanical properties, anti-corrosion/fatigue properties, which has been widely used in various fields, such as aerospace, automation, civil engineering, architecture, ocean engineering, etc. However, synthetic fibers have some disadvantage, such as environmental pollution, renewable raw materials and expensive price. In contrast, natural fibers can overcome some advantages of traditional synthetic fibers and attract more interest in different fields in recent years. Therefore, it is suggested that the authors should firstly analyze the advantages and disadvantages of synthetic fibers in the first pagagraph, and further summary some development potential of natural fibers composites. The following are some latest research on the properties of synthetic fiber composites in different fields. International Journal of Fatigue, 2020, 134: 105480. Composite Structures, 2019, 229: 111427. Materials and Structures, 2020, 53: 73.

3. At the end of the introduction, it is suggested that the authors further emphasize the contribution and innovation of the current research work and some major problems to be solved.

4. In part 2.1: Does the use of flame retardant affect the interfacial properties of fibers? On the other hand, will the addition of flame retardant to epoxy increase the water absorption performance? Did the authors consider the above factors when selecting materials?

5. In part 2.2.1 (Ageing), please provide the basis for selecting the hygrothermal ageing conditions, including temperature, humidity and maximum exposure time. Is it not enough for the maximum exposure time of 30 days? In addition, please further indicate the test interval of water uptake.

6. It can be found in Fig. 2 that the moisture absorption content increases with the increase of water activity at 70°C. Please explain the reason. As known, the number of pores in the composite material is certain. Therefore, at the same temperature, the composite material reaches the water absorption equilibrium state after the water value completely fills the pore. What is the reason for the difference in the water absorption equilibrium content in Figure 2? The similar condition also applies to Figure 4b.

7. In Fig. 9, why does the tensile strength increase first and then decrease with the increase of the equilibrium water absorption? This is inconsistent with common cognition.

8. In general, after the water absorption, the brittleness of epoxy resin will be strengthened due to the reaction between resin matrix and water molecules. Why does the elongation at break of the composite gradually increase with the increase of the equilibrium water absorption in figure 11? Please provide relevant explanations.

9. The conclusions should be further improved, including 3-4 key points, according to the current results and findings.

Reviewer 2 Report

The paper seeks to introduce an approach “Effect of hygrothermal ageing on the mechanical and fire properties of a flame retarded flax fibres/epoxy composite”. However, the authors should consider improving upon the quality to further highlight and emphasize.

1.     The word “sensivity” is wrongly spelt. Consider replacing it with the right word as “sensitivity”.

2.    Based on the understanding of what constitute an abstract, consider adding one or two lines highlighting the significance of the study.

3.    Put space between each variable and its corresponding unit. For instance, in lines 48 and 53, instead of 62.5 % and 70 °C, the author represented them as 62.5 % and 70 °C. Consider correcting all throughout the article.

4.    The introduction needs to be improved by relating to the mechanics of the studied materials and their mechanical characteristics. The references to be included are: 10.1177/0021998318790093, 10.1016/j.polymertesting.2017.09.009, 10.1002/app.46770, 10.1016/j.compstruct.2021.114698, 10.1177/0731684417727143 10.3390/polym14132662, 10.1016/j.porgcoat.2022.107015.

5.    There are some English mistakes which the author has to read the whole manuscript and accordingly correct them.

6.    Tabulate all the materials used with their physical and chemical properties.

7.    In section 2.2.1, how many samples were immersed in the deionized water?

8.    In the SEM characterization, what is the scale bar used?

9.    Figures are poorly inserted and therefore blurry. Consider reinserting as a text but not as a snapshot.

Round 2

Reviewer 1 Report

It is suggested to accept this paper.